# Potentially inappropriate prescribing in older adults with advanced chronic kidney disease

**Amber O. Molnar** [1,2,3]*, **Sarah Bota**[3], **Nivethika Jeyakumar**[3], **Eric McArthur**[3], **Marisa Battistella**[4], **Amit X. Garg**[5], **Manish M. Sood**[6,7], **K. Scott Brimble**[1]

**1** Division of Nephrology, Department of Medicine, McMaster University, Hamilton, Ontario, Canada, **2** Department of Health Research Methods, Evidence, and Impact, McMaster University, Hamilton, Ontario, Canada, **3** ICES, Toronto, Ontario, Canada, **4** University Health Network/Leslie Dan Faculty of Pharmacy, University of Toronto, Toronto, Ontario, Canada, **5** Division of Nephrology, Department of Medicine, Western University, London, Ontario, Canada, **6** Division of Nephrology, Department of Medicine, University of Ottawa, Ottawa, Canada, **7** Epidemiology, Ottawa Hospital Research Institute, Ottawa, Ontario, Canada

* amolnar@stjosham.on.ca

## Abstract

### Background

Older adults with chronic kidney disease (CKD) are at heightened risk for polypharmacy. We examined potentially inappropriate prescribing in this population and whether introducing pharmacists into the ambulatory kidney care model was associated with improved prescribing practices.

### Methods

Retrospective cohort study using linked administrative databases. We included patients with an eGFR $\leq$30 mL/min/1.73 m$^2$ $\geq$66 years of age followed in multidisciplinary kidney clinics in Ontario, Canada (n = 25,016 from 28 centres). The primary outcome was the absence of a statin prescription or the receipt of a potentially inappropriate prescription defined by the American Geriatric Society Beers Criteria$^®$ and a modified Delphi panel that identified key drugs of concern in CKD. We calculated the crude cumulative incidence and incidence rate for the primary outcome and used change-point regression to determine if a change occurred following pharmacist introduction.

### Results

There were 6,007 (24%) and 16,497 patients (66%) not prescribed a statin and with $\geq$1 potentially inappropriate prescription, respectively. The rate of potentially inappropriate prescribing was 125.6 per 100 person-years and was higher in more recent years. The change-point regression analysis included 2,275 patients from two centres. No immediate change was detected at pharmacist introduction, but potentially inappropriate prescribing was increasing pre-pharmacist introduction, and this rising trend was reversed post-pharmacist introduction. The incidence of potentially inappropriate prescribing still remained high post-pharmacist introduction.

**Data Availability Statement:** The data set from this study is held securely in coded form at ICES. While data sharing agreements prohibit ICES from making the data set publicly available, access may

be granted to those who meet pre-specified criteria for confidential access, available at https://www.ices.on.ca/DAS. The full data set creation plan and underlying analytic code are available from the authors upon request, understanding that the programs may rely upon coding templates or macros that are unique to ICES.

**Funding:** This study was conducted with the support of Cancer Care Ontario through funding provided by the Government of Ontario (awarded to AOM). The sponsor had no role in the study design, conduct, data analysis or manuscript preparation. This study was supported by ICES, which is funded by an annual grant from the Ontario Ministry of Health and Long-Term Care (MOHLTC). This study was completed at the ICES Western site, where core funding is provided by the Academic Medical Organization of Southwestern Ontario, the Schulich School of Medicine and Dentistry, Western University, and the Lawson Health Research Institute. Amber O. Molnar receives salary support from the KRESCENT Foundation and the McMaster Department of Medicine. Manish M Sood is supported by the Jindal Research Chair for the Prevention of Kidney Disease.

**Competing interests:** Manish M. Sood has received grant funding from Otsuka and speaker fees from Astrazeneca unrelated to this study. All other authors have no conflicts to declare. This does not alter our adherence to PLOS ONE policies on sharing data and materials.

## Conclusions

Potentially inappropriate prescribing practices were common. Incorporating pharmacists into the kidney care model may improve prescribing practices. The role of pharmacists in the ambulatory kidney care team warrants further investigation in a randomized controlled trial.

## Introduction

Older adults ($\geq$65 years) with advanced chronic kidney disease (CKD) (estimated glomerular filtration rate (eGFR) <30 mL/min/1.73 m$^2$) are a growing patient population [1]. The risk of adverse drug reactions is increased in this population due to polypharmacy as well as altered drug pharmacokinetics and pharmacodynamics caused by age-related changes, altered nutritional state, and reduced kidney clearance [2–4]. Therefore, dose adjustment and enhanced monitoring or complete discontinuation and avoidance are required for several medications in order to prevent drug accumulation that may lead to toxicity [2, 5–9]. The prevalence of potentially inappropriate prescribing in patients with CKD in the ambulatory setting varies from 13% to 96%, depending on the patient population and how CKD and inappropriate prescribing are defined [10–13]. There are limited data on how best to reduce potentially inappropriate prescribing in patients with advanced CKD. Studies suggest that involving pharmacists in the care of patients with CKD improves prescribing practices [10]. However, most studies have focused on prescribing in the acute care hospital setting or hemodialysis population, or have involved recommendations from a community pharmacist and have not examined the impact of pharmacists as part of the ambulatory kidney care team [10, 14–16]. With this in mind, we conducted a retrospective cohort study to examine potentially inappropriate prescribing in older patients with advanced CKD followed in multidisciplinary kidney clinics and examined if inappropriate prescribing was reduced following the introduction of pharmacists into the kidney clinics. We anticipated a high rate of inappropriate prescribing and that introducing pharmacists into the kidney clinics would be associated with improved prescribing practices.

## Materials and methods

### Design and setting

We conducted a retrospective cohort study using administrative healthcare databases linked via unique encoded identifiers and analyzed at ICES in Ontario, Canada. The study was conducted according to a pre-specified protocol. The use of data in this project was authorized under section 45 of Ontario's Personal Health Information Protection Act, which does not require review by a Research Ethics Board. The reporting of this study follows the Reporting of Studies Conducted Using Observational Routinely Collected Health Data (RECORD) guidelines for observational studies (Appendix A in S1 File) [17].

We included patients followed in multidisciplinary kidney clinics across the province of Ontario, Canada. Adults ($\geq$18 years of age) with advanced CKD (eGFR <30 mL/min/1.73 m$^2$) are referred for care in the clinics at the discretion of their treating nephrologist. The care provided in multidisciplinary clinics across the province of Ontario is not standardized; therefore, the role and availability of interdisciplinary healthcare professionals may vary across clinics. The care team may include the following healthcare professionals: nephrologist, nurse

practitioner, nurse, dietitian, social worker, pharmacist and diabetes nurse educator. The focus of multidisciplinary kidney clinics is to provide education, manage CKD complications, prevent CKD progression, and prepare patients for kidney replacement therapy. Pharmacists may see patients at each clinic visit to perform medication reconciliation and provide education. Any identified medication concerns would be discussed with the treating nephrologist, which may result in changes to the medication regimen by the nephrologist or a written or verbal communication with the physician prescribing any drug(s) of concern. Pharmacists also serve as a resource for nephrologists regarding drug dosing and drug interactions when prescribing new medications.

## Data sources

Details regarding databases used for the study are outlined in Appendix B in S1 File. Patients followed in multidisciplinary kidney clinics were identified using the Ontario Renal Reporting System (ORRS). The Ontario Health Insurance Plan (OHIP) database, the Canadian Institute for Health Information Discharge Abstract Database (CIHI-DAD) and ORRS were used to identify patients with a prior history of maintenance dialysis, or a history of a kidney transplant (exclusion criteria). Baseline laboratory data were determined using the Ontario Laboratory Information System (OLIS). Serum creatinine concentrations from OLIS, which were all measured using the isotope dilution mass spectroscopy–traceable enzymatic method, were used to calculate eGFR using the Chronic Kidney Disease Epidemiology (CKD-EPI) equation [18]. Demographics and vital status information were obtained from the Ontario Registered Persons Database. Diagnostic and procedure information from all hospitalizations were determined using the CIHI-DAD and CIHI-Same Day Surgery database. Diagnostic information from emergency room visits was determined using the CIHI-National Ambulatory Care Reporting System (NACRS). Medication data were obtained from the Ontario Drug Benefit Plan database, which contains highly accurate records of all outpatient prescriptions dispensed to patients ≥65 years [19]. Whenever possible, we defined patient characteristics and outcomes using validated codes (Appendix C in S1 File).

## Study cohort

We included patients with active follow-up in multidisciplinary kidney clinics from April 1, 2011 to March 31, 2017. The first clinic visit date within this time period was the cohort entry date (index date). We excluded individuals less than 66 years of age, without an eGFR measurement in the year prior to the index date, or with a history of dialysis or kidney transplant. To ensure only patients with advanced CKD were included in the cohort, patients with an eGFR >30 mL/min/1.73 m$^2$ (based on the most recent value prior to the index date) were excluded. We assessed comorbidities in the 5 years prior to the index date and albuminuria and eGFR (taking the most recent value) in the year prior to the index date. Baseline medication use was determined based on prescriptions dispensed in the 120 days prior to the index date.

## Potentially inappropriate prescribing

Potentially inappropriate prescribing was defined by the absence of a statin prescription from a patient's index date to the end of follow-up or receipt of a potentially inappropriate prescription at any point from the index date to the end of the follow-up. Patients were followed until they experienced a censoring event (discharge or withdrawal from the multidisciplinary kidney clinic, death, kidney transplant, or maintenance dialysis initiation) or maximum follow-up occurred (March 31, 2018). The absence of a statin prescription was considered potentially

inappropriate given that the Kidney Disease Improving Global Outcomes (KDIGO) Clinical Practice Guideline For Lipid Management in CKD recommends that all adults ≥50 years of age with an eGFR <60 mL/min/1.73 m$^2$ be treated with a statin or statin/ezetimibe combination (Grade 1A evidence) [20]. Potentially inappropriate prescriptions (see Appendix D in S1 File) were defined by the American Geriatric Society Beers Criteria® [5], (medications contraindicated or to be prescribed with caution in older persons), and a modified Delphi panel that identified key medications of concern in CKD [21]. Additional medications of concern in CKD and kidney dosing guidelines were obtained from the Compendium of Pharmaceuticals and Specialties (CPS) and Micromedex [22, 23]. Potentially inappropriate prescriptions were further classified into the following categories: medications of concern in CKD, medications of concern in older patients, medications recommended to be avoided in patients with an eGFR <15 mL/min/1.73 m$^2$ (examined in patients with a baseline eGFR <15 mL/min/1.73 m$^2$, n = 5,689), and medications with clear dosing guidelines dispensed above the recommended dose for an eGFR <30 mL/min/1.73 m$^2$ (Appendix E in S1 File). Non-steroidal anti-inflammatory medications were not examined due to the common non-prescription use of these medications, which is not captured in our databases.

## Statistical analysis

We determined the crude cumulative incidence of potentially inappropriate prescribing by dividing the total number of patients with one or more potentially inappropriate prescriptions (fill date on or after the index date) or with absence of a statin prescription throughout the follow-up by the total number of patients. The potentially inappropriate prescribing rate per 100 person-years was calculated by dividing the total person-years of potentially inappropriate prescribing (based on time supplied ≥1 potentially inappropriate prescription or time with no statin prescription) by the total person-years of follow-up. Potentially inappropriate prescribing rates were stratified by age (66-<80 and ≥80), sex, and index year and incidence rate ratios were calculated for each subgroup of interest. Rates per 100 person-years were calculated for each potentially inappropriate prescription category. The crude cumulative incidence for each potentially inappropriate medication of interest was calculated (total number of patients with ≥1 prescription for each medication of interest divided by the total number of patients) to determine the most commonly prescribed medications.

We used change-point regression with monthly intervals to determine if there was a difference in the risk of potentially inappropriate prescribing before and after pharmacist introduction into multidisciplinary kidney clinics. Change-point regression analysis allows for an intervention effect to be studied over time, accounting for prior trends in the outcome, seasonality, and correlation between time points. It also allows the assessment of whether an intervention has an immediate effect in the outcome of interest or if there is an effect over time post-intervention [24]. There were two centres that had introduced a pharmacist into the clinic early in the accrual period, providing sufficient pre- and post-pharmacist data. The first centre had introduced a pharmacist in November 2013; providing a pre-pharmacist time period from April 1, 2011 to October 31, 2013 and post-pharmacist time period from November 1, 2013 to March 31, 2018. The second centre had introduced a pharmacist in May 2014; providing a pre-pharmacist time period from April 1, 2011 to April 30, 2014 and post-pharmacist time period from May 1, 2014 to March 31, 2018. Data were pooled from the two centres and arranged into monthly intervals relative to the pharmacist start date. Absolute standardized differences were used to compare baseline characteristics pre- and post-pharmacist introduction; a value ≥0.1 was considered a significant imbalance between the two time periods. The proportion of patients with potentially inappropriate prescribing during each monthly interval

was calculated. Change in the risk of potentially inappropriate prescribing was estimated using linear regression. We tested for the presence of autocorrelation using the Durbin-Watson statistic and if there was evidence of autocorrelation, we included an autocorrelation term at lag 1 in the model. A sensitivity analysis examining the impact of pharmacist introduction on the mean number of potentially inappropriate prescriptions per patient during each monthly interval was performed (absence of statin prescription excluded from this analysis). We conducted all analyses using SAS version 9.4 (SAS Institute, Cary, NC).

## Results

### Baseline characteristics

Once all exclusion criteria were applied, 25,016 patients from 28 multidisciplinary kidney clinics were included (S1 Fig). The mean (standard deviation, SD) age was 78 (7.4) years and 56% were male. Patients were on a mean (SD) of 10 (4.8) medications at baseline. The mean (SD) baseline eGFR was 19.9 (6.0) mL/min/1.73 $m^2$; 23% of the cohort had an eGFR <15 mL/min/1.73 $m^2$ (Table 1).

**Table 1. Baseline characteristics.**

| Characteristic | All patients N = 25,016 |
|---|---|
| **Demographics** | |
| Age, mean (SD) | 78 (7.4) |
| Sex (male), n (%) | 14,000 (56.0) |
| Rural[a], n (%) | 2,913 (11.6) |
| Index Year, n (%) | |
| 2011 | 5,611 (22.4) |
| 2012 | 2,033 (8.1) |
| 2013 | 3,737 (14.9) |
| 2014 | 4,786 (19.1) |
| 2015 | 4,712 (18.8) |
| 2016 | 3,387 (13.5) |
| 2017 | 750 (3.0) |
| **Comorbidities[b]** | |
| Atrial fibrillation, n (%) | 3,689 (14.7) |
| Chronic obstructive pulmonary disease, n (%) | 2,160 (8.6) |
| Congestive heart failure, n (%) | 5,519 (22.1) |
| Diabetes, n (%) | 16,077 (64.3) |
| Hypertension, n (%) | 23,659 (94.6) |
| Myocardial infarction, n (%) | 2,336 (9.3) |
| Peripheral vascular disease, n (%) | 1,060 (4.2) |
| **Kidney Function[c]** | |
| Serum creatinine (μmol/L), mean (SD) | 254.2 (97.2) |
| eGFR (mL/min/1.73 m2), mean (SD) | 19.9 (6.0) |
| eGFR <15 (mL/min/1.73 m2), n (%) | 5,689 (22.7) |
| Urine albumin creatinine ratio (mg/mmol), mean (SD)[d] | 96.6 (157.6) |
| **Medication Use[e]** | |
| Number of prescribed medications, mean (SD) | 10 (4.8) |
| Colchicine, n (%) | ≤5 (0.0) |
| Lithium, n (%) | 53 (0.2) |
| Spironolactone, n (%) | 1,653 (6.6) |

(*Continued*)

**Table 1.** (Continued)

| Characteristic | All patients N = 25,016 |
|---|---|
| Methotrexate, n (%) | 88 (0.4) |
| Fibrates, n (%) | 556 (2.2) |
| Glyburide, n (%) | 662 (2.6) |
| Metformin, n (%) | 2,319 (9.3) |
| Sodium glucose transporter-2 inhibitors, n (%) | 14 (0.1) |
| Ciprofloxacin, n (%) | 1,497 (6.0) |
| Levofloxacin, n (%) | 462 (1.8) |
| Nitrofurantoin, n (%) | 605 (2.4) |
| Baclofen, n (%) | 124 (0.5) |
| Valacyclovir or acyclovir, n (%) | 143 (0.6) |
| Digoxin, n (%) | 0 (0.0) |
| Pregabalin, n (%) | 645 (2.6) |
| Gabapentin, n (%) | 931 (3.7) |
| Morphine, n (%) | 211 (0.8) |
| Codeine, n (%) | 2,533 (10.1) |
| Duloxetine | 17 (0.1) |
| Peripheral alpha-blockers, n (%) | 3,974 (15.9) |
| Alpha agonists, n (%) | 468 (1.9) |
| Tricyclic anti-depressants, n (%) | 791 (3.2) |
| Paroxetine, n (%) | 276 (1.1) |
| Benzodiazepines, n (%) | 2,936 (11.7) |
| Proton pump inhibitors, n (%) | 10,471 (41.9) |
| Metoclopramide, n (%) | 128 (0.5) |
| Skeletal muscle relaxants, n (%) | 0 (0.0) |
| First generation antihistamines, n (%) | 8 (0.0) |
| Anti-arrhythmic drugs, n (%) | 950 (3.8) |
| Anti-psychotics, n (%) | 253 (1.0) |
| Direct oral anticoagulants, n (%) | 533 (2.1) |
| Statins, n (%) | 17,595 (70.3) |

[a]Rural is defined as residing in a location with a population of ≤10,000 individuals.

[b]Comorbidities in the 5 years prior to index date were considered.

[c]Laboratory measurements in the year prior to index date were considered, using the most recent value. eGFR was determined using the CKD-EPI equation.

[d]Missing values, n = 7,986 (32%)

[e]Prescriptions dispensed in the 120 days prior to index date were considered.

*In accordance with ICES privacy policies, cell sizes less than or equal to five cannot be reported.

## Potentially inappropriate prescribing

The cumulative incidence of potentially inappropriate prescribing was 22,504 out of 25,016 (90%) patients over a median (interquartile range, IQR) follow-up of 2.0 (1.1 to 3.2) years [absence of a statin prescription: 6,007 (24%); ≥1 potentially inappropriate prescription: 16,497 (66%)]. The overall rate of potentially inappropriate prescribing was 125.6 per 100 person-years, calculated by dividing 72,453 total person years of potentially inappropriate prescribing by 57,707 total person years of follow-up. The potentially inappropriate prescribing rate did not differ by age category (incidence rate ratio, IRR, 0.99, 95% confidence interval, CI,

**Table 2. Potentially inappropriate prescribing rates.**

| | N | Total person years of potentially inappropriate prescribing | Total person years of follow up | Potentially inappropriate prescribing rate per 100 person-years | IRR (95% CI) | IRR p value |
|---|---|---|---|---|---|---|
| Total cohort | 25,016 | 72,453 | 57,707 | 125.6 | | |
| **Age (years)** | | | | | | |
| 66-<80 | 14,490 | 43,534 | 34,576 | 125.9 | Reference | 0.35 |
| ≥80 | 10,526 | 28,919 | 23,131 | 125.0 | 0.99 (0.98, 1.01) | |
| **Sex** | | | | | | |
| Male | 14,000 | 37,315 | 31,431 | 118.7 | Reference | <0.0001 |
| Female | 11,016 | 35,138 | 26,276 | 133.7 | 1.13 (1.11, 1.14) | |
| **Index year** | | | | | | |
| 2011 | 5,611 | 25,630 | 20,865 | 122.8 | Reference | |
| 2012 | 2,033 | 6,470 | 5,198 | 124.5 | 1.01 (0.99, 1.04) | 0.34 |
| 2013 | 3,737 | 11,344 | 8,835 | 128.4 | 1.05 (1.02, 1.07) | <0.0001 |
| 2014 | 4,786 | 12,516 | 9,997 | 125.2 | 1.02 (1.00, 1.04) | 0.08 |
| 2015 | 4,712 | 10,205 | 7,945 | 128.5 | 1.05 (1.02, 1.07) | 0.0001 |
| 2016 | 3,387 | 5,425 | 4,206 | 129.0 | 1.05 (1.02, 1.08) | 0.001 |
| 2017 | 750 | 864 | 662 | 130.4 | 1.06 (0.99, 1.14) | 0.08 |
| **Potentially inappropriate prescription categories** | | | | | | |
| Medications of concern in CKD | 25,016 | 12,156 | 57,707 | 21.1 | | |
| Medications of concern in older patients | 25,016 | 42,808 | 57,707 | 74.2 | | |
| Medications to be avoided at an eGFR <15 mL/min/1.73 m2[a] | 5,689 | 336 | 9,791 | 3.4 | | |
| Medications dispensed above the recommended dose for an eGFR <30 mL/min/1.73 m2 | 25,016 | 47 | 57,707 | 0.1 | | |

[a]Examined in sub-group of patients with a baseline eGFR <15 mL/min/1.73 m$^2$.

Abbreviations: CKD, chronic kidney disease, eGFR: estimated glomerular filtration rate, IRR: incidence rate ratio.

0.98 to 1.01), but was higher in female patients (IRR 1.13, 95% CI 1.11 to 1.14) and in more recent index years. The most commonly prescribed potentially inappropriate prescription category was medications of concern in older patients (74.2 per 100 person-years), followed by medications of concern in CKD (21.1 per 100 person-years). Medications to be avoided at an eGFR <15 mL/min/1.73 m$^2$ and medications dispensed above the recommended dose for an eGFR <30 mL/min/1.73 m$^2$ were prescribed at low rates (3.4 and 0.1 per 100 person-years, respectively) (Table 2).

Potentially inappropriate medications with the highest cumulative incidence (determined by ≥1 prescription per patient) were proton pump inhibitors (PPI) (consecutive use >8 weeks) (40%), codeine (30%), peripheral alpha-blockers (23%), ciprofloxacin (any dose) (21%), and benzodiazepines (21%). With respect to indications for peripheral alpha-blocker use, 636 (11%) patients with an alpha-blocker prescription had a prior diagnosis of benign prostatic hyperplasia within five years prior to the first prescription. Pregabalin and

gabapentin (commonly prescribed for neuropathic pain) were prescribed to 1,835 (7%) and 2,135 (9%) patients, respectively. Among those prescribed pregabalin, 264 (14%) filled at least one prescription with a dose >150 mg per day. Among those prescribed gabapentin, 504 (23%) filled at least one prescription with a dose >700 mg per day. The cumulative incidence for each medication in the categories of medications recommended to be avoided in patients with an eGFR <15 mL/min/1.73 $m^2$ and medications dispensed above the recommended dose for an eGFR <30 mL/min/1.73 $m^2$ are detailed in S1 and S2 Tables, respectively. Primary care physicians were responsible for most potentially inappropriate prescriptions, but were also responsible for most statin prescriptions (Table 3).

### Impact of pharmacists in multidisciplinary kidney clinics on potentially inappropriate prescribing

There were 2,275 patients from two centres included in the change point regression analysis. There were minor differences in baseline characteristics pre- and post-pharmacist introduction (S3 Table). No change in polypharmacy was observed from the first to last study interval (mean number of medications = 10). The proportion of patients with potentially inappropriate prescribing was compared pre- and post-pharmacist introduction. No immediate change at pharmacist introduction was detected (p = 0.14), but the slope pre-pharmacist introduction was positive, indicating a rising proportion of individuals with potentially inappropriate prescribing over the months prior to pharmacist introduction (p<0.001). The slope changed to negative post-pharmacist introduction, indicating that the rise in potentially inappropriate prescribing was reversed and a slight decline over the months post-pharmacist introduction was observed (p<0.001). However, the incidence of potentially inappropriate prescribing still remained high (Fig 1, S4 Table). When the category of medications of concern in CKD was examined, a rising trend of potentially inappropriate prescriptions was seen pre-pharmacist introduction (p = 0.003), which continued immediately post-pharmacist introduction (p<0.001), but a significant decline was then noted post-pharmacist introduction (p = 0.003) (Fig 2, S5 Table). For medications of concern in older patients, potentially inappropriate prescriptions were increasing pre-pharmacist introduction (p = 0.024), and an immediate (p<0.001), sustained decline was noted post-pharmacist introduction (p = 0.0003) (Fig 3, S6 Table). A sensitivity analysis that examined the mean number of potentially inappropriate prescriptions per patient also showed that a rising trend of potentially inappropriate prescriptions was reversed post-pharmacist introduction (S2 Fig, S7 Table).

### Discussion

In this retrospective cohort study of 25,016 older patients with advanced CKD, we found that polypharmacy was common (mean of 10 medications per patient) and that potentially inappropriate prescribing occurred in 90% of patients at some point over the follow-up (almost

**Table 3. Medical specialty of prescribers.**

| Physician specialty | Potentially inappropriate prescriptions (%) | Potentially inappropriate prescriptions for medications of concern in CKD (%) | Statin prescriptions (%) |
|---|---|---|---|
| Primary care | 75.6 | 72.9 | 77.3 |
| Nephrology | 9.8 | 8.9 | 9.6 |
| Other | 8.6 | 11.8 | 7.5 |
| Missing | 6.0 | 6.4 | 5.6 |

Abbreviations: CKD: chronic kidney disease.

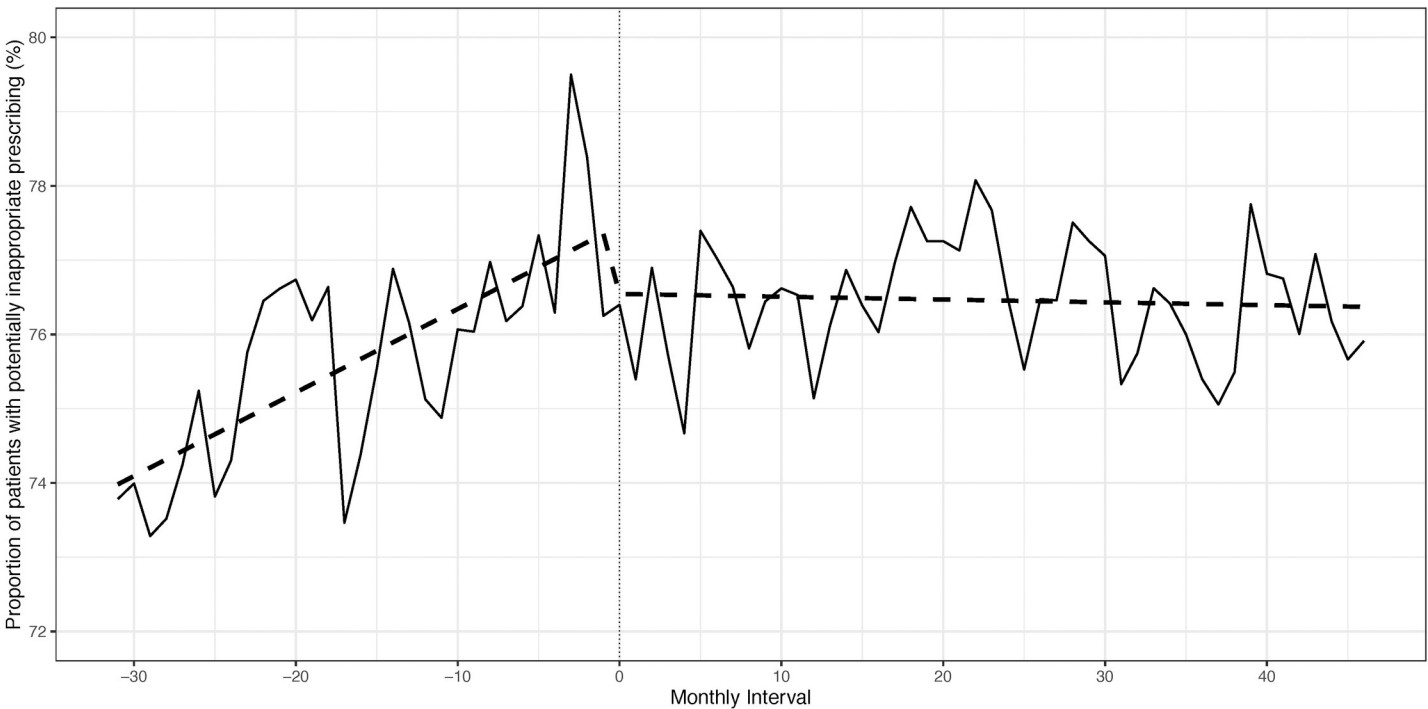

**Fig 1. Proportion of patients with potentially inappropriate prescribing pre- and post-pharmacist introduction.**

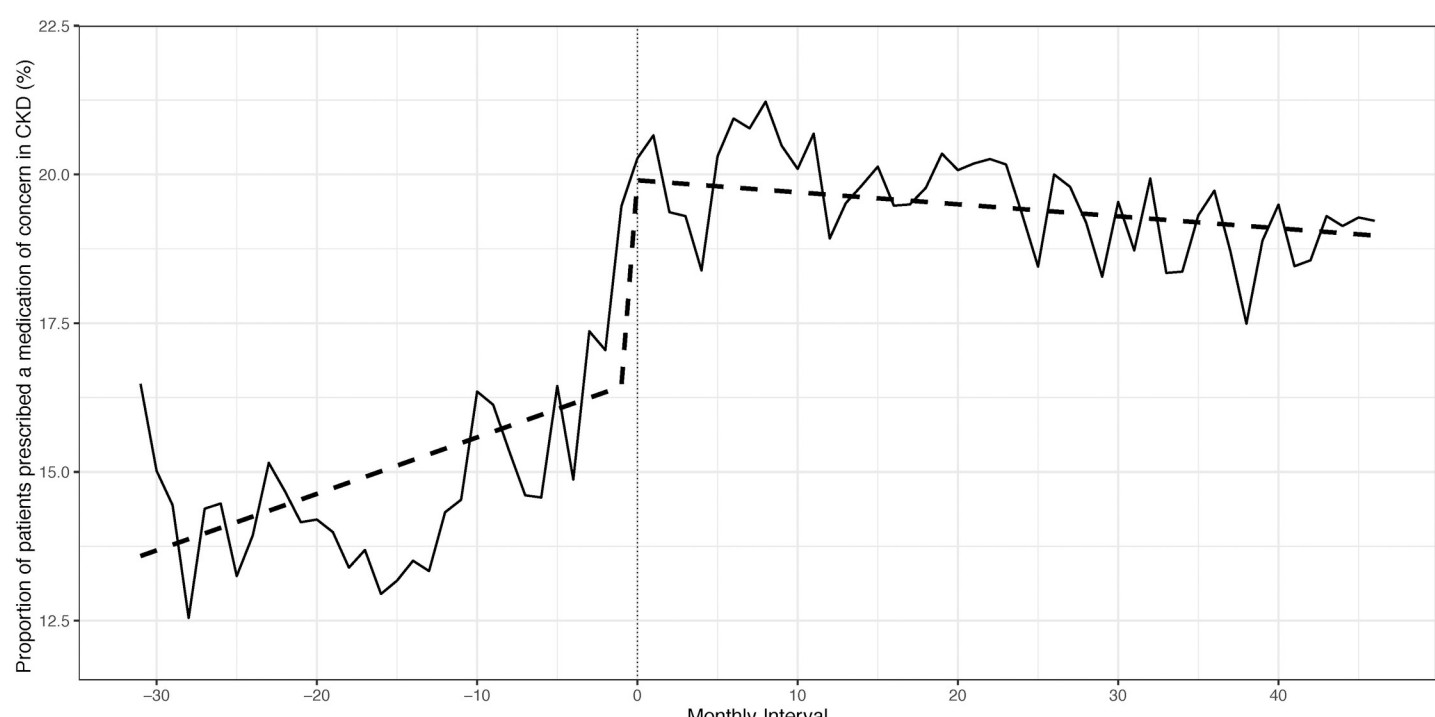

**Fig 2. Proportion of patients prescribed a medication of concern in CKD pre- and post-pharmacist introduction.**

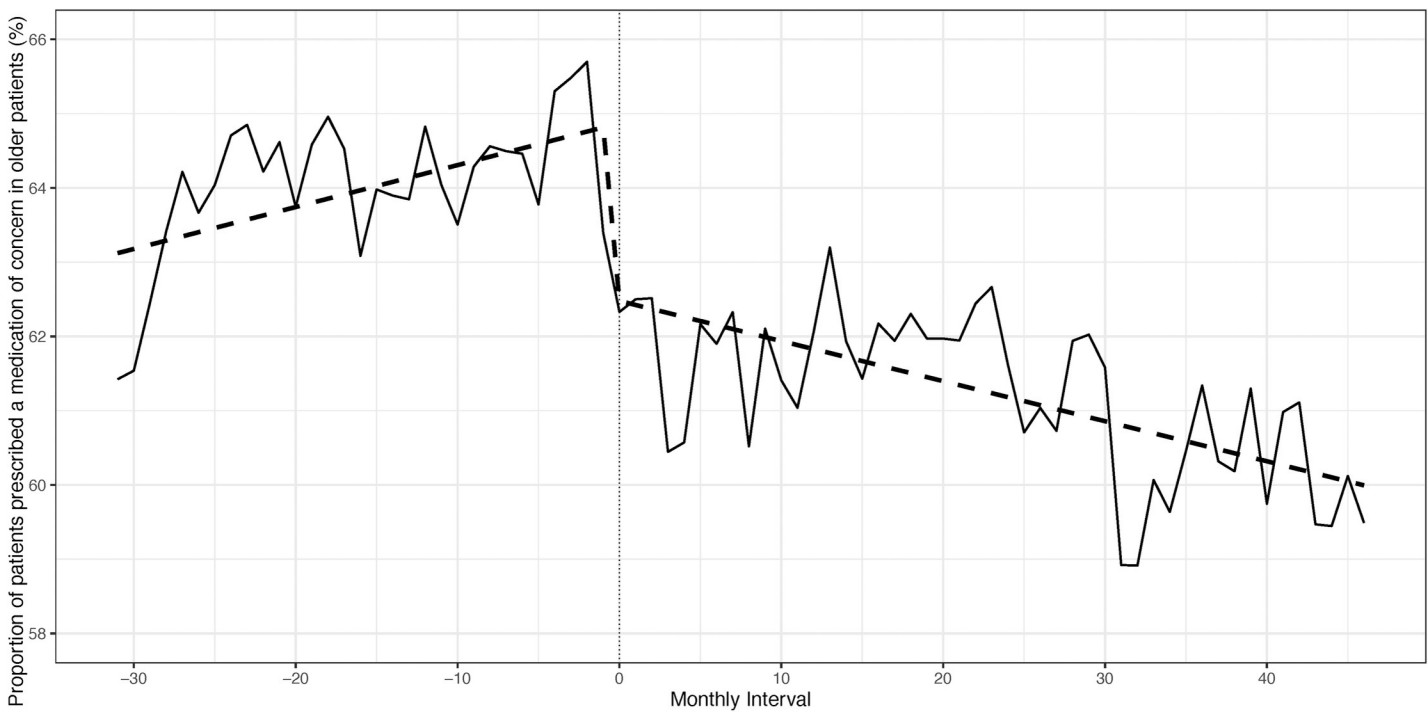

**Fig 3. Proportion of patients prescribed a medication of concern in older patients pre- and post-pharmacist introduction.**

one quarter due to the absence of a statin prescription). Pharmacist presence in multidisciplinary kidney clinics was associated with a significant reduction in potentially inappropriate prescribing; suggesting that the inclusion of pharmacists as part of the ambulatory kidney care team improves prescribing practices in a population that is at high risk for adverse drug reactions. However, it should be noted that a very modest reduction was observed, and potentially inappropriate prescribing still remained common post pharmacist introduction.

Our findings are similar to prior studies examining prescribing in patients with kidney disease [10, 25]. The reported prevalence of potentially inappropriate prescribing in patients with CKD is quite variable, depending on the patient population studied and how potentially inappropriate prescribing is defined [10]. We found a higher potentially inappropriate prescribing rate in more recent years, which is consistent with prior studies [26]. We also found a higher potentially inappropriate prescribing rate in women, which has been previously reported [27]. Medications of concern in older patients were prescribed at the highest rate, primarily driven by chronic PPI prescriptions. Chronic PPI use was the most common potentially inappropriate prescribing practice (40%), which is not surprising given the dramatic increase in long-term PPI prescribing over the past two decades and the fact that PPIs are the second most commonly prescribed drug in Canada [26, 28]. Prolonged PPI administration is of concern due to the association with an increased risk of *Clostridium difficile* colitis, pneumonia, fractures and, more recently, CKD [29–33]. The high frequency of chronic PPI use suggests that clinicians are not attempting to de-prescribe PPIs after a course of at least 4 weeks and no further symptoms, as recommended by guidelines [34, 35]. Peripheral alpha-blockers, also of concern in older patients, were commonly prescribed. This may be due to the high prevalence of resistant hypertension in patients with advanced CKD [36], which often requires the addition of less desirable anti-hypertensive medications to achieve blood pressure targets. The low prevalence of benign prostatic hyperplasia found in patients prescribed these agents suggests that

they were primarily prescribed for hypertension. While alpha-blocker use may be appropriate in certain older patients, prescribers must be aware of the heightened risk for orthostatic hypotension causing falls [5, 37]. The opioid codeine, which is a high-risk medication in older patients, was commonly prescribed. Codeine requires dose adjustment in CKD and has an unpredictable response depending on the rate of drug metabolism [38, 39]. Of further concern is the frequent prescribing of benzodiazepines and, to a lesser degree, gapabentin and pregabalin, which are all associated with an increased risk of death when co-administered with opioids [40–43].

Absolutely contraindicated medications or doses clearly outside the recommended range for eGFR were fortunately prescribed at relatively low rates. Fluoroquinolones were the most common medication class dispensed above recommended doses and were commonly prescribed. Antibiotics have been previously reported as medications at high risk of inappropriate prescribing in patients with CKD. Interestingly, automated eGFR reporting has not been found to improve antibiotic prescribing practices [44, 45]. Patients with kidney failure (eGFR $<15$ mL/min/1.73 m$^2$) filled prescriptions for medications contraindicated at very low levels of kidney function, such as metformin, fibrates, baclofen and glyburide. These types of prescriptions are most concerning because they place patients at highest risk for serious adverse drug reactions.

Nearly 25% of the cohort was not prescribed a statin despite CKD guidelines providing a strong recommendation to prescribe these agents to all patients with CKD above the age of 50 [20]. One potential reason for the lack of a statin prescription could be side effects prompting discontinuation. Unfortunately, this information was not available in our databases. However, the reported prevalence of statin-related side effects ranges from 1–10% [46], which is much lower than 25%. One other reason may be therapeutic nihilism on the part of prescribers. Our cohort consisted of older patients with advanced CKD; a patient population typically excluded from cardiovascular therapeutic trials [47, 48].

When prescribing over multiple monthly intervals was compared, the introduction of pharmacists into multidisciplinary kidney clinics was associated with a reduction in potentially inappropriate prescribing, and a sustained effect was noted. A significant, immediate reduction in potentially inappropriate prescribing at the time of pharmacist introduction was not observed in all analyses. We did not necessarily anticipate an immediate impact given that first-time visits with a pharmacist occurred at the time of routine clinic visits and would therefore be expected to occur over several months. Consistent with our findings, a beneficial impact of pharmacists on prescribing practices has been previously demonstrated in other clinical settings and patient populations with kidney disease [10]. Although not examined in this study, reducing potentially inappropriate prescribing is clinically important since this should reduce adverse drug reactions, pill burden, and costs for the patient as well as the healthcare system in jurisdictions with publically funded drug plans [49, 50]. It is however important to note that even with the introduction of pharmacists, we still found that the incidence of potentially inappropriate prescribing remained high, suggesting that a multi-faceted intervention is needed to address this issue. A successful intervention would likely need to incorporate primary care physicians since we found that they were responsible for most potentially inappropriate prescriptions and most statin prescriptions.

The generalizability of our findings is increased by the inclusion of a large, multi-centre cohort with universal drug coverage. However, our study has limitations worth noting. Our databases lacked information on why potentially inappropriate prescribing occurred; individual patient tolerability of potentially inappropriate medications, side effects and any therapeutic benefits were also not available. Our list of potentially inappropriate prescribing practices was based on expert opinion and published guidelines; however, we acknowledge that the

evidence to support the avoidance or dose reduction of many included drugs is limited. Also, the best equation to estimate kidney function for the purposes of drug adjustment or avoidance continues to be controversial. The National Kidney Disease Education Program indicates that equations which express results in mL/min/1.73 m$^2$ or mL/min are both appropriate for this purpose. In this study, we estimated the glomerular filtration rate using the CKD-EPI equation, which when <30 mL/min/1.73 m$^2$, would also generally identify a patient with a Cockcroft–Gault creatinine clearance <30 mL/min [51]. Our finding that pharmacist presence in multidisciplinary kidney clinics was associated with improved prescribing practices may not be causal since the intervention was not randomly assigned. Other factors occurring at the same time that pharmacists were introduced may have led to the observed reduction in potentially inappropriate prescribing. However, the finding is strengthened by the fact that the reduction in potentially inappropriate prescribing was noted across two centres that each introduced pharmacists at different time periods (November 2013 and May 2014). Lastly, the role and practice of pharmacists at each clinic were not standardized.

## Conclusions

Potentially inappropriate prescribing was common in this cohort of older patients with advanced CKD. Our findings demonstrate the important need to improve outpatient-prescribing practices in this high-risk population and that including pharmacists in the delivery of ambulatory kidney care might improve prescribing. The role and impact of pharmacists as part of the ambulatory kidney care team warrants further investigation in a randomized controlled trial.

## Supporting information

**S1 Fig. Cohort creation.**
(TIFF)

**S2 Fig. Mean number of potentially inappropriate prescriptions per patient pre- and post-pharmacist introduction.**
(TIFF)

**S1 Table. Cumulative incidence of medications recommended to be avoided in patients with an eGFR <15 mL/min/1.73 m2.**
(DOCX)

**S2 Table. Cumulative incidence of medications dispensed above the recommended dose for an eGFR <30 mL/min/1.73 m2.**
(DOCX)

**S3 Table. Baseline characteristics pre-and post-pharmacist introduction.**
(DOCX)

**S4 Table. Change point regression analysis examining proportion of patients with potentially inappropriate prescribing pre-and post-pharmacist introduction.**
(DOCX)

**S5 Table. Change point regression analysis examining proportion of patients prescribed at least one medication of concern in CKD pre-and post-pharmacist introduction.**
(DOCX)

**S6 Table. Change point regression analysis examining proportion of patients prescribed at least one medication of concern in the elderly pre-and post-pharmacist introduction.**
(DOCX)

**S7 Table. Change point regression analysis examining the mean number of potentially inappropriate prescriptions per patient pre-and post-pharmacist introduction.**
(DOCX)

**S1 File. Appendices.**
(DOCX)

## Acknowledgments

The research was conducted by members of the ICES Kidney, Dialysis and Transplantation team, at the ICES Western facility. Parts of this material are based on data and information compiled and provided by the Ontario Ministry of Health and Long-Term Care (MOHLTC), Canadian Institute for Health Information (CIHI) and Cancer Care Ontario (CCO). The analyses, conclusions, opinions and statements expressed herein are solely those of the authors and do not reflect those of the funding or data sources; no endorsement is intended or should be inferred. Parts of this material are based on data and/or information compiled and provided by CIHI. However, the analyses, conclusions, opinions and statements expressed in the material are those of the authors, and not necessarily those of CIHI. Parts of this material are based on data and information provided by CCO. The opinions, results, view, and conclusions reported in this paper are those of the authors and do not necessarily reflect those of CCO. No endorsement by CCO is intended or should be inferred. *We* thank IMS Brogan Inc. for use of their Drug Information Database.

## Author Contributions

**Conceptualization:** Amber O. Molnar, Marisa Battistella, K. Scott Brimble.

**Data curation:** Amber O. Molnar, Eric McArthur.

**Formal analysis:** Amber O. Molnar, Eric McArthur.

**Funding acquisition:** Amber O. Molnar, K. Scott Brimble.

**Investigation:** Amber O. Molnar.

**Methodology:** Amber O. Molnar, Eric McArthur.

**Project administration:** Amber O. Molnar, Sarah Bota, Nivethika Jeyakumar.

**Supervision:** Amber O. Molnar.

**Writing – original draft:** Amber O. Molnar, K. Scott Brimble.

**Writing – review & editing:** Amber O. Molnar, Sarah Bota, Nivethika Jeyakumar, Eric McArthur, Marisa Battistella, Amit X. Garg, Manish M. Sood.

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
