## [Decision Letter · Decision Letter 0]

4 May 2020

PONE-D-20-07703

Potentially inappropriate prescribing in older adults with advanced CKD

PLOS ONE

Dear Dr Molnar,

Thank you for submitting your manuscript to PLOS ONE. After careful consideration, we feel that it has merit but does not fully meet PLOS ONE’s publication criteria as it currently stands. Therefore, we invite you to submit a revised version of the manuscript that addresses the points raised during the review process.

Thank you for the submission of this manuscript and I apologise for the delay in the review process. I have recommended minor corrections in accordance with the reviewers. Both reviewer 1 and 3 requested clarification regarding the statistical analyses and on review, the time-series analysis should be paid some attention. There are several approaches to interrupted time-series analysis so please explain the rationale for the approach taken any limitations to the approach. Due to the complexity of the analysis, revisions may require statistical review.

We would appreciate receiving your revised manuscript by Jun 18 2020 11:59PM. To enhance the reproducibility of your results, we recommend that if applicable you deposit your laboratory protocols in protocols.io, where a protocol can be assigned its own identifier (DOI) such that it can be cited independently in the future. For instructions see: http://journals.plos.org/plosone/s/submission-guidelines#loc-laboratory-protocols

We look forward to receiving your revised manuscript.

Kind regards,

Carl Richard Schneider, BN, BPharm (Hon), PhD

Academic Editor

PLOS ONE

Journal Requirements:

https://doi.org/10.1093/ndt/gfz167

In your revision ensure you cite all your sources (including your own works), and quote or rephrase any duplicated text outside the Methods section. Further consideration is dependent on these concerns being addressed.

'This study was supported by the ICES Western and Ottawa sites. ICES is funded by an annual grant from the Ontario Ministry of Health and Long-Term Care (MOHLTC). Core funding for ICES Western is provided by the Academic Medical Organization of Southwestern Onta 342 rio (AMOSO), the Schulich School of Medicine and Dentistry (SSMD), Western University, and the Lawson Health Research Institute (LHRI). The research was conducted by members of the ICES Kidney, Dialysis and Transplantation team, at the ICES Ottawa and Western facilities, who are supported by a grant from the Canadian Institutes of Health Research (CIHR). The opinions, results and conclusions are those of the authors and are independent from the funding sources. No endorsement by ICES, AMOSO, SSMD, LHRI, CIHR, or the MOHLTC is intended or should be inferred.

Parts of this material are based on data and/or information compiled and provided by CIHI. However, the analyses, conclusions, opinions and statements expressed in the material are those of the authors, and not necessarily those of CIHI. Amber O Molnar receives salary support from the KRESCENT Foundation and the McMaster Department of Medicine. Manish M Sood is supported by the Jindal Research Chair for the

Prevention of Kidney Disease.'

'Yes. This study was conducted with the support of Cancer Care Ontario through

funding provided by the Government of Ontario (awarded to AOM;

https://www.cancercareontario.ca/en). The sponsor had no role in the study design,

conduct, data analysis or manuscript preparation.'

6. Thank you for stating the following in your Competing Interests section: 

'I have read the journal's policy and the authors of this manuscript have the following

competing interests: Manish M. Sood has received grant funding from Otsuka and

speaker fees from Astrazeneca unrelated to this study. All other authors have no

conflicts to declare.'

Additional Editor Comments (if provided):

Reviewers' comments:

Reviewer's Responses to Questions

**Comments to the Author**

1. Is the manuscript technically sound, and do the data support the conclusions?

Reviewer #1: Yes

Reviewer #2: Yes

Reviewer #3: Partly

2. Has the statistical analysis been performed appropriately and rigorously? 

Reviewer #1: Yes

Reviewer #2: Yes

Reviewer #3: Yes

3. Have the authors made all data underlying the findings in their manuscript fully available?

Reviewer #1: Yes

Reviewer #2: Yes

Reviewer #3: Yes

4. Is the manuscript presented in an intelligible fashion and written in standard English?

Reviewer #1: Yes

Reviewer #2: Yes

Reviewer #3: Yes

5. Review Comments to the Author

Reviewer #1: Comments to authors:

This study is a retrospective analysis of linked administrative databases from Canada examining the incidence of inappropriate prescribing in older adults with advanced CKD and evaluating the impact of including pharmacists as part of the multidisciplinary team in the ambulatory kidney care model on inappropriate prescribing.

The study is relevant, was conducted by a highly qualified team, and the manuscript reads well. I applaud the authors for following the Reporting of Studies Conducted Using Observational Routinely Collected Health Data (RECORD) guidelines for observational studies.

I have a few suggestions to improve the manuscript, which I hope the authors find helpful. Thank you for the opportunity to review this paper.

Introduction

Line 60: there is a more recent systematic review pertaining to the role of pharmacists in CKD that you may want to cite. PMID: 30963447

Methods

Lines 80-95: it seems that the information about the clinics would make more sense to be included under “Setting”

Line 92: is there information regarding pharmacist recommendations and acceptance rate by physicians?

Line 66: what was the rationale for picking equal or greater than 66 years-old instead of equal or greater than 65 years-old to define older adults?

Line 121: the authors refer to measures of “healthcare utilization”. What measures do the authors mean and what were they used for?

Line 126-148: I suggest rearranging this section in the following manner: 1) lines 127-129; 2) lines 139-143; 3) lines 132-136; 4) lines 143-148; 5) end of line 136 through 138: 6) lines 130-131.

Line 132: the authors used the 2015 version of the Beers criteria. Would you anticipate any changes to the results had you used the 2019 version of the Beers criteria (PMID: 30693946)?

Lines 151-158: Cumulative incidence calculation – can the authors provide more information about the numerator and denominator used to calculate cumulative incidence to help guide the reader? What were considered ‘new cases’ and what was the ‘number of individuals free of disease at the beginning of time period’ per the definition of cumulative incidence?

Along the same lines, the authors state that “The number of days each patient had potentially inappropriate prescribing […] and the total follow-up days for each patient were used to determine the potentially inappropriate prescribing rate per 100 person-years.” How does the ‘number of days’ give a rate in ‘person-years’. On Table 2, it seems that the rate of potentially inappropriate prescribing was calculated by dividing ‘total person years of potentially inappropriate prescribing’ by ‘total person years of follow-up’, which I can understand. I think it will help the reader if you clarify what the numerator and denominator are for every calculation performed.

Line 196: please state the statistical level used in the analysis.

Results

Line 193: please state clearly what the calculated cumulative incidence is as well as numerator and denominator.

Line 196: The overall rate of potentially inappropriate prescribing was 125.6 per 100 person-years, calculated by dividing 72,453 total person years of potentially inappropriate prescribing by 57,707 total person years of follow-up. Please clarify the n and the follow-up period (is it 2011-2017?) used to calculate the latter rates.

Line 224: “No immediate change at pharmacist introduction was detected” – it looks like it became significant after pharmacist introduction (p<0.001) and the difference between slopes was also significant (p<0.001) per information on Supplementary Table S2. Can that be considered more than just a trend? The authors summarize the findings stating that there was a "significant reduction in potentially inappropriate prescribing" (Lines 246-247).

References

Reference #48: there seems to be an issue with the author name.

Tables and Figures

Table 1: Suggest including what variables are presented as mean (SD) and what variables are presented as n(%) on the table and not as a footnote. It is confusing as it stands.

Table 3: in the fourth column, I suggest clarifying that this means statins prescribed

Figure 1: this figure is hard to read and the color coding is not apparent because the figure is in black and white

Reviewer #2: I would like to commend the authors for their work.

It is well written and argued paper, which I am happy to accept in its current form.

I was just curious if there is a reason why females had more rates of potentially inappropriate medication. Does it have anything to do with the higher probability of some drugs which are inappropriate in females?

Otherwise, I am happy to accept this work as is.

Reviewer #3: Thank you for the opportunity to review the manuscript. The manuscript is well written and adds to the literature for support to include pharmacists as part of the MD team in CKD.

However, few points below for clarification

• Suggest adding the definition for older adults in the introduction

• What was the rationale for involving only <30ml patients?

• Why was the only use of statin examined? There are other potential therapies that may be under used.

• What about the use of ESAs? ESA can be inappropriate especially with regards to the Hb rise?

• How many patients were conservatively managed? Or under renal supportive care team given that a significant proportion is very old

• Suggest adding further details on the 24% who were not prescribed statins. How many of them had coexisting CVD

• Suggest adding some details on the use of certain meds. For example, prazosin can be a very useful agent to reduce BP in patients with CKD and is used frequently in Australia. Furthermore, this population is older. Although less preferred it could be used for BPH. Gabapentin and pregabalin are also used with appropriate dose management

• Suggest adding a supplementary table with all meds prescribed inappropriately with dosage considerations

• Suggest a stat review. I am not clear on how the pharmacists involvement conclusion has been performed.

6. PLOS authors have the option to publish the peer review history of their article (what does this mean?). If published, this will include your full peer review and any attached files.

Reviewer #1: Yes: Teresa M Salgado

Reviewer #2: No

Reviewer #3: No

---

## [Author Response · Author response to Decision Letter 0]

15 Jul 2020

Dear Dr Molnar,

Thank you for submitting your manuscript to PLOS ONE. After careful consideration, we feel that it has merit but does not fully meet PLOS ONE’s publication criteria as it currently stands. Therefore, we invite you to submit a revised version of the manuscript that addresses the points raised during the review process.

Thank you for the submission of this manuscript and I apologise for the delay in the review process. I have recommended minor corrections in accordance with the reviewers. Both reviewer 1 and 3 requested clarification regarding the statistical analyses and on review, the time-series analysis should be paid some attention. There are several approaches to interrupted time-series analysis so please explain the rationale for the approach taken any limitations to the approach. Due to the complexity of the analysis, revisions may require statistical review.

Response to Editors’ comments:

We have added the following rationale to the statistical analysis section: “Change-point regression analysis allows for an intervention effect to be studied over time, accounting for prior trends in the outcome, seasonality, and correlation between time points. It also allows the assessment of whether an intervention has an immediate effect in the outcome of interest or if there is an effect over time post-intervention.”

Reference: Wagner, A. K., Soumerai, S. B., Zhang, F., & Ross‐Degnan, D. (2002). Segmented regression analysis of interrupted time series studies in medication use research. Journal of clinical pharmacy and therapeutics, 27(4), 299-309.

Limitations of the change point regression analysis include the fact that it assumes a linear trend in the outcome over time points, and it aggregates individual-level data, which does not allow for adjustment of individual-level characteristics. However, there did not appear to be obvious non-linearity in our observed trends, and we would not expect confounders to change substantially over the study window of approximately 6 years.

Journal Requirements:

Whilst you may use any professional scientific editing service of your choice, PLOS has partnered with both American Journal Experts (AJE) and Editage to provide discounted services to PLOS authors. Both organizations have experience helping authors meet PLOS guidelines and can provide language editing, translation, manuscript formatting, and figure formatting to ensure your manuscript meets our submission guidelines. To take advantage of our partnership with AJE, visit the AJE website (http://secure-web.cisco.com/1591UxBoAQ5ywtRJ47Xo0nlWQuOqVDzYDKRTkkBnPJxwZENpv87T2Sp5GNbfZT1Bx_xEXXaqKdii2FWSA0S5tZjpvcoBvLdiDqLsiEv7GJ4eHFGd30Gyh3-h4QFIo-C9h1c68Qd5i26GdhBUK3pinRjqTFQzFFUmlN7kQ-APOb0yIm3m25wh-JwSfAL5Gmo_5b-RmY8jN6yfORGp5Q250v9RxaPYT2rCbh2q-9H_HcTsry08rh-aolM91oIq9d4orXqQoPsq1uQZorzQJcMH8kw/http%3A%2F%2Flearn.aje.com%2Fplos%2F) for a 15% discount off AJE services. To take advantage of our partnership with Editage, visit the Editage website (www.editage.com) and enter referral code PLOSEDIT for a 15% discount off Editage services. If the PLOS editorial team finds any language issues in text that either AJE or Editage has edited, the service provider will re-edit the text for free.

Sarah Bota thoroughly proofread the manuscript prior to submission.

https://doi.org/10.1093/ndt/gfz167

In your revision ensure you cite all your sources (including your own works), and quote or rephrase any duplicated text outside the Methods section. Further consideration is dependent on these concerns being addressed.

Can the editors please clarify which text is overlapping outside the Methods section? The paper with overlapping text is also one of our publications conducted using similar databases so there will be text from the Methods section that is overlapping. In reviewing both manuscripts, we were unable to find the overlapping text outside of the Methods section that is referred to.

This is the standard language we use pertaining to data requests for all studies conducted using ICES data: “The data set from this study is held securely in coded form at ICES. While data sharing agreements prohibit ICES from making the data set publicly available, access may be granted to those who meet pre-specified criteria for confidential access, available at www.ices.on.ca/DAS. The full data set creation plan and underlying analytic code are available from the authors upon request, understanding that the programs may rely upon coding templates or macros that are unique to ICES.”

ICES receives a vast majority of its information from Ontario’s publicly funded health care system. The Personal Health Information Protection Act (PHIPA) protects individuals’ privacy by outlining rules for the collection, use and disclosure of health information. ICES is designated as a Prescribed Entity by the Information and Privacy Commissioner (IPC) of Ontario under section 45 of PHIPA and is required to implement controls on the access to our data repositories. One such control is to administer access of information on a project-by-project basis. 

If further information is required please contact:

Pamela Seto

ICES Legal Counsel

Pamela.Seto@ices.on.ca

'This study was supported by the ICES Western and Ottawa sites. ICES is funded by an annual grant from the Ontario Ministry of Health and Long-Term Care (MOHLTC). Core funding for ICES Western is provided by the Academic Medical Organization of Southwestern Ontario (AMOSO), the Schulich School of Medicine and Dentistry (SSMD), Western University, and the Lawson Health Research Institute (LHRI). The research was conducted by members of the ICES Kidney, Dialysis and Transplantation team, at the ICES Ottawa and Western facilities, who are supported by a grant from the Canadian Institutes of Health Research (CIHR). The opinions, results and conclusions are those of the authors and are independent from the funding sources. No endorsement by ICES, AMOSO, SSMD, LHRI, CIHR, or the MOHLTC is intended or should be inferred.

Parts of this material are based on data and/or information compiled and provided by CIHI. However, the analyses, conclusions, opinions and statements expressed in the material are those of the authors, and not necessarily those of CIHI. Amber O Molnar receives salary support from the KRESCENT Foundation and the McMaster Department of Medicine. Manish M Sood is supported by the Jindal Research Chair for the

Prevention of Kidney Disease.'

'Yes. This study was conducted with the support of Cancer Care Ontario through

funding provided by the Government of Ontario (awarded to AOM;

https://secure-web.cisco.com/1-mnyuSXeJgYvQJIKGa84nC6PpnwKNbRy8EY7CIM5cKnjidNBzev9ZaOx5yfOgbxnpVaSDu6VT-9009gLZHB8v-Sibv8GqnqXdv58mquQ43tx-Wdw3jr55fWIEo7a2seyzl_w-wvGH3m69ktg2uJQhAMUSugyJD-8X2wUyz9Am9RESfM8_wDjLely7FXgcgUakWhZo5zGDqx-cV9ikr6Lep59i36WhvVIrSEXF0gxmbeoj7ieMz_5_tpIOVhE2J8F/https%3A%2F%2Fwww.cancercareontario.ca%2Fen). The sponsor had no role in the study design,

conduct, data analysis or manuscript preparation.'

Please update the funding statement to read: “This study was conducted with the support of Cancer Care Ontario through funding provided by the Government of Ontario (awarded to AOM). The sponsor had no role in the study design, conduct, data analysis or manuscript preparation. This study was supported by ICES, which is funded by an annual grant from the Ontario Ministry of Health and Long-Term Care (MOHLTC). This study was completed at the ICES Western site, where core funding is provided by the Academic Medical Organization of Southwestern Ontario, the Schulich School of Medicine and Dentistry, Western University, and the Lawson Health Research Institute. Amber O. Molnar receives salary support from the KRESCENT Foundation and the McMaster Department of Medicine. Manish M Sood is supported by the Jindal Research Chair for the Prevention of Kidney Disease.”

The Acknowledgments section has been updated accordingly in the manuscript.

6. Thank you for stating the following in your Competing Interests section: 

'I have read the journal's policy and the authors of this manuscript have the following

competing interests: Manish M. Sood has received grant funding from Otsuka and

speaker fees from Astrazeneca unrelated to this study. All other authors have no

conflicts to declare.'

Additional Editor Comments (if provided):

Reviewers' comments:

Reviewer's Responses to Questions

Comments to the Author

1. Is the manuscript technically sound, and do the data support the conclusions?

Reviewer #1: Yes

Reviewer #2: Yes

Reviewer #3: Partly

2. Has the statistical analysis been performed appropriately and rigorously? 

Reviewer #1: Yes

Reviewer #2: Yes

Reviewer #3: Yes

3. Have the authors made all data underlying the findings in their manuscript fully available?

Reviewer #1: Yes

Reviewer #2: Yes

Reviewer #3: Yes

4. Is the manuscript presented in an intelligible fashion and written in standard English?

Reviewer #1: Yes

Reviewer #2: Yes

Reviewer #3: Yes

5. Review Comments to the Author

Reviewer #1: Comments to authors:

This study is a retrospective analysis of linked administrative databases from Canada examining the incidence of inappropriate prescribing in older adults with advanced CKD and evaluating the impact of including pharmacists as part of the multidisciplinary team in the ambulatory kidney care model on inappropriate prescribing.

The study is relevant, was conducted by a highly qualified team, and the manuscript reads well. I applaud the authors for following the Reporting of Studies Conducted Using Observational Routinely Collected Health Data (RECORD) guidelines for observational studies.

I have a few suggestions to improve the manuscript, which I hope the authors find helpful. Thank you for the opportunity to review this paper.

Introduction

Line 60: there is a more recent systematic review pertaining to the role of pharmacists in CKD that you may want to cite. PMID: 30963447

Thank you for pointing this out. This reference has now been added to the introduction.

Methods

Lines 80-95: it seems that the information about the clinics would make more sense to be included under “Setting”

The description about the clinics and the role of pharmacists has now been moved to the Design and Setting section.

Line 92: is there information regarding pharmacist recommendations and acceptance rate by physicians?

Unfortunately the databases used for the study do not have this information available.

Line 66: what was the rationale for picking equal or greater than 66 years-old instead of equal or greater than 65 years-old to define older adults?

We have medication data available for patients 65 years and older. The purpose of selecting 66 years and older is to allow for a consistent determination of baseline medication use prior to study entry in case patients enter the study at 65 years of age.

Line 121: the authors refer to measures of “healthcare utilization”. What measures do the authors mean and what were they used for?

Thank you for pointing out this inconsistency. This statement regarding measures of healthcare utilization has now been removed from the manuscript.

Line 126-148: I suggest rearranging this section in the following manner: 1) lines 127-129; 2) lines 139-143; 3) lines 132-136; 4) lines 143-148; 5) end of line 136 through 138: 6) lines 130-131.

This section has been rearranged as suggested.

Line 132: the authors used the 2015 version of the Beers criteria. Would you anticipate any changes to the results had you used the 2019 version of the Beers criteria (PMID: 30693946)?

We appreciate this comment. We used the 2015 version given that the planning of our study and cohort inclusion dates (2011-2017) pre-dated the 2019 version of the Beers criteria. In comparing the 2 versions, we do not anticipate any changes to the results had we used the 2019 version.

Lines 151-158: Cumulative incidence calculation – can the authors provide more information about the numerator and denominator used to calculate cumulative incidence to help guide the reader? What were considered ‘new cases’ and what was the ‘number of individuals free of disease at the beginning of time period’ per the definition of cumulative incidence?

Along the same lines, the authors state that “The number of days each patient had potentially inappropriate prescribing […] and the total follow-up days for each patient were used to determine the potentially inappropriate prescribing rate per 100 person-years.” How does the ‘number of days’ give a rate in ‘person-years’. On Table 2, it seems that the rate of potentially inappropriate prescribing was calculated by dividing ‘total person years of potentially inappropriate prescribing’ by ‘total person years of follow-up’, which I can understand. I think it will help the reader if you clarify what the numerator and denominator are for every calculation performed.

We have added in further details to the statistical analysis section as to how the calculations were performed: “We determined the cumulative incidence of potentially inappropriate prescribing by dividing the total number of patients with one or more potentially inappropriate prescriptions (fill date on or after the index date) or with absence of a statin prescription throughout the follow-up by the total number of patients. The potentially inappropriate prescribing rate per 100 person-years was calculated by dividing the total person-years of potentially inappropriate prescribing by the total person-years of follow-up… The crude cumulative incidence for each potentially inappropriate medication of interest was calculated (total number of patients with ≥1 prescription for a particular medication throughout the follow-up divided by the total number of patients) to determine the most commonly prescribed medications.”

Line 196: please state the statistical level used in the analysis.

The overall rate of potentially inappropriate prescribing was 125.6 per 100 person-years.

We are unclear as to what specifically the reviewer is referring to with this comment. We have overall added further details to the Results and to Table 2, which hopefully addresses any concerns that the reviewer had.

Results

Line 193: please state clearly what the calculated cumulative incidence is as well as numerator and denominator.

The statement has been revised to the following: “The cumulative incidence of potentially inappropriate prescribing was 22,504 out of 25,016 (90%) patients over a median (interquartile range, IQR) follow up of 2.0 (1.1-3.2) years [absence of a statin prescription: 6,007 (24%); ≥1 potentially inappropriate prescription: 16,497 (66%)].”

Line 196: The overall rate of potentially inappropriate prescribing was 125.6 per 100 person-years, calculated by dividing 72,453 total person years of potentially inappropriate prescribing by 57,707 total person years of follow-up. Please clarify the n and the follow-up period (is it 2011-2017?) used to calculate the latter rates.

Further details regarding n for each subgroup have been added to Table 2. The follow-up period for all calculations was from study inclusion (index date) (which could occur for a patient any time between Apr 1, 2011 until March 31, 2017) until a censoring event or maximum follow up (March 31, 2018) occurred. These details are specified in the Study cohort and Potentially inappropriate prescribing sections of the Methods section. 

Line 224: “No immediate change at pharmacist introduction was detected” – it looks like it became significant after pharmacist introduction (p<0.001) and the difference between slopes was also significant (p<0.001) per information on Supplementary Table S2. Can that be considered more than just a trend? The authors summarize the findings stating that there was a "significant reduction in potentially inappropriate prescribing" (Lines 246-247).

Thank you for this comment. We have revised the text at line 224 to the following: “No immediate change at pharmacist introduction was detected, but the slope pre-pharmacist introduction was positive, indicating a rising proportion of individuals with potentially inappropriate prescribing over the months prior to pharmacist introduction. The slope changed to negative post-pharmacist introduction, indicating that the rise in potentially inappropriate prescribing was reversed and a slight decline over the months post-pharmacist introduction was observed.” We believe that the use of the word trend was misleading and was not meant to reference “a statistical trend”. We have also revised the supplemental tables to include further information in the footnotes that should assist with the interpretation of the p values presented.

References

Reference #48: there seems to be an issue with the author name.

This has been corrected (now reference #51).

Tables and Figures

Table 1: Suggest including what variables are presented as mean (SD) and what variables are presented as n(%) on the table and not as a footnote. It is confusing as it stands.

This has been changed as suggested.

Table 3: in the fourth column, I suggest clarifying that this means statins prescribed

This has been clarified as suggested.

Figure 1: this figure is hard to read and the color coding is not apparent because the figure is in black and white

Thank you for pointing this out. We realized that Fig 1 was confusing and it has now been removed.

Reviewer #2: I would like to commend the authors for their work.

It is well written and argued paper, which I am happy to accept in its current form.

I was just curious if there is a reason why females had more rates of potentially inappropriate medication. Does it have anything to do with the higher probability of some drugs which are inappropriate in females?

We agree that this finding is interesting and has been previously demonstrated (https://www.ncbi.nlm.nih.gov/pmc/articles/PMC5942474/). We have the following statement to the discussion: “We also found a higher potentially inappropriate prescribing rate in women, which has been reported in prior studies.26”

We agree that it is likely due to a higher probability of prescribing for certain symptoms or conditions, such as mood or sleep disturbance. This could be related to the fact that women are more likely to discuss these concerns with their physician and seek treatment.

Otherwise, I am happy to accept this work as is.

Reviewer #3: Thank you for the opportunity to review the manuscript. The manuscript is well written and adds to the literature for support to include pharmacists as part of the MD team in CKD.

However, few points below for clarification

• Suggest adding the definition for older adults in the introduction

This has been added.

• What was the rationale for involving only <30ml patients?

Patients are required to have an eGFR <30 mL/min/1.73 m2 in order to be followed in the multidisciplinary kidney clinic.

• Why was the only use of statin examined? There are other potential therapies that may be under used.

We were particularly interested in statin prescriptions since it is a therapy that requires very little monitoring and typically once prescribed, is continued indefinitely with little to no periods of interruption. Also, no specific conditions beyond CKD are needed for a statin to be recommended.

For other therapies, such as ASA or angiotensin receptor blockers (ARB) or angiotensin converting enzyme inhibitors (ACEi), other conditions on top of CKD would be needed for a prescription to be recommended (i.e. diabetes or cardiovascular disease). While we could look for those conditions in our databases, we would then have to account for diagnostic code inaccuracies and concerns when commenting on appropriate use. For ACEi and ARB use in this patient population, there is also the issue of inconsistent prescribing, dose titration or discontinuation due to hyperkalemia or AKI, and we would not have blood pressure measurement data or universal proteinuria data.

• What about the use of ESAs? ESA can be inappropriate especially with regards to the Hb rise?

We fully agree with this comment. Unfortunately, the use of ESAs is not captured in our databases and hemoglobin measurements were not universally available.

• How many patients were conservatively managed? Or under renal supportive care team given that a significant proportion is very old

Unfortunately, our databases at the time of this study were not capturing this information in a reliable manner.

• Suggest adding further details on the 24% who were not prescribed statins. How many of them had coexisting CVD

While we agree this may add insight as to why clinicians may be less aggressive about prescribing a statin to these patients, the CKD Lipid Guidelines do not require a prior diagnosis of CVD to recommend statin use given the high risk of cardiovascular events in patients with CKD. Our objective was to examine whether or not CKD on its own prompted clinicians to prescribe a statin.

• Suggest adding some details on the use of certain meds. For example, prazosin can be a very useful agent to reduce BP in patients with CKD and is used frequently in Australia. Furthermore, this population is older. Although less preferred it could be used for BPH. Gabapentin and pregabalin are also used with appropriate dose management

The following details have now been added to the Results section:

With respect to indications for peripheral alpha-blocker use, only 636 (11%) patients with an alpha-blocker prescription had a diagnosis of benign prostatic hyperplasia in the five years prior to the prescription. Pregabalin and gabapentin (commonly prescribed for neuropathic pain) were prescribed to 1835 (7%) and 2135 (9%) patients, respectively. Among those prescribed pregabalin, 264 (14%) filled at least one prescription with a dose >150 mg per day. Among those prescribed gabapentin, 504 (23%) filled at least one prescription with a dose >700 mg per day.”

• Suggest adding a supplementary table with all meds prescribed inappropriately with dosage considerations

Supplemental Tables 1 and 2 now detail the cumulative incidence of each medication in the categories of medications recommended to be avoided in patients with an eGFR <15 mL/min/1.73 m2 and medications dispensed above the recommended dose for an eGFR <30 mL/min/1.73 m2.

• Suggest a stat review. I am not clear on how the pharmacists involvement conclusion has been performed.

We have added further details to the Methods section regarding the rationale and limitations of time series analysis.

---

## [Decision Letter · Decision Letter 1]

5 Aug 2020

Potentially inappropriate prescribing in older adults with advanced chronic kidney disease

PONE-D-20-07703R1

Dear Dr. Molnar,

We’re pleased to inform you that your manuscript has been judged scientifically suitable for publication and will be formally accepted for publication once it meets all outstanding technical requirements.

Kind regards,

Carl Richard Schneider, BN, BPharm (Hon), PhD

Academic Editor

PLOS ONE

Additional Editor Comments (optional):

Thank you for the careful revision. There are a couple of additional minor comments by Reviewer 1 that you may wish to consider for the final manuscript.

Reviewers' comments:

Reviewer's Responses to Questions

**Comments to the Author**

1. If the authors have adequately addressed your comments raised in a previous round of review and you feel that this manuscript is now acceptable for publication, you may indicate that here to bypass the “Comments to the Author” section, enter your conflict of interest statement in the “Confidential to Editor” section, and submit your "Accept" recommendation.

Reviewer #1: All comments have been addressed

2. Is the manuscript technically sound, and do the data support the conclusions?

Reviewer #1: Yes

3. Has the statistical analysis been performed appropriately and rigorously? 

Reviewer #1: Yes

4. Have the authors made all data underlying the findings in their manuscript fully available?

Reviewer #1: Yes

5. Is the manuscript presented in an intelligible fashion and written in standard English?

Reviewer #1: Yes

6. Review Comments to the Author

Reviewer #1: Thank you for addressing all my comments. I think you did a really nice job with the manuscript, congratulations!

A couple minor comments:

Page 5, line 74 - please spell out ICES

Page 19, lines 328-334 - suggest presenting some data here to support the claim that the slopes differed before and after pharmacists introduction in the clinics.

7. PLOS authors have the option to publish the peer review history of their article (what does this mean?). If published, this will include your full peer review and any attached files.

Reviewer #1: **Yes: **Teresa M Salgado

---

## [Editor Report · Acceptance letter]

10 Aug 2020

PONE-D-20-07703R1 

Potentially inappropriate prescribing in older adults with advanced chronic kidney disease 

Dear Dr. Molnar:

I'm pleased to inform you that your manuscript has been deemed suitable for publication in PLOS ONE. Congratulations! Your manuscript is now with our production department. 

Kind regards, 

on behalf of

Dr. Carl Richard Schneider 

Academic Editor

PLOS ONE